# Hospital care does not meet the communication needs of patients with hearing loss: A qualitative study of patient experiences

Alex DeBusschere[1], Meaghan Lunney[1], Sonja Reid[2], Nancy Verdin[2,3], Shannan Love[1], Stephanie Thompson[4], David Nicholas[5], Tiffany Boulton[6], Kara Schick-Makaroff[7], Lorienne Jenstad[8], Sharon Straus[9], Jayna Holroyd-Leduc[1], Maoliosa Donald[1], Patti-Jo Sullivan[10], Tanis Howarth[10], Julie Evans[10], Marcello Tonelli[1]*

1 Department of Medicine, University of Calgary, Calgary, Alberta, Canada, 2 Patient Partner, Calgary, Alberta, Canada, 3 Patient Partner, Red Deer, Alberta, Canada, 4 Department of Medicine, University of Alberta, Edmonton, Alberta, Canada, 5 Faculty of Social Work, University of Calgary, Calgary, Alberta, Canada, 6 Department of Community Health Sciences, University of Calgary, Calgary, Alberta, Canada, 7 Faculty of Nursing, University of Alberta, Edmonton, Alberta, Canada, 8 School of Audiology and Speech Sciences, University of British Columbia, Vancouver, British Columbia, Canada, 9 Department of Medicine, University of Toronto, Toronto, Ontario, Canada, 10 Alberta Health Services, Edmonton, Alberta, Canada

* Tonelli.admin@ucalgary.ca

## Abstract

### Background

Good communication is essential for high quality healthcare. People with hearing loss face communication challenges during health encounters, which may compromise their experience with care and outcomes, especially in noisy or stressful acute care facilities. Tools and strategies to facilitate two-way communication with these patients and other members of their healthcare team may help address this gap. This study describes the communication-related experiences of patients with hearing loss in Alberta hospitals, which can help inform future strategies in this setting.

### Methods

Drawing on qualitative description, we conducted focus groups and individual interviews with people with hearing loss who had a recent hospital experience in Alberta, Canada. Focus group and interview transcripts were abductively coded and analyzed, guided by established communication frameworks to explore experiences and opportunities for change.

### Results

Fourteen people participated in 3 focus groups and 3 interviews. Overall, participants perceived hospitals as not meeting their communication needs. We identified 3

**Data availability statement:** Data cannot be shared publicly due to risks of identifying patient information. Requests to access to data may be made by qualified researchers to the University of Calgary Conjoint Health Research Ethics Board who can field data inquiries from fellow researchers: email: chreb@ucalgary.ca; tel: (403) 220-2297.

**Funding:** The study was supported by a CIHR project grant to MT (award number 202309PJT). The funding organization had no role in the design and conduct of the study; collection, management, analysis, and interpretation of the data; preparation, review, or approval of the manuscript; and decision to submit the manuscript for publication.

**Competing interests:** The authors have declared that no competing interests exist.

themes: 1) *Hearing loss is an invisible disability*; 2) *Communication is a team effort*; 3) *Every patient has different needs in different situations*.

## Conclusions

Patients with hearing loss experience communication gaps while in hospital, possibly related to the invisibility and stigma of hearing loss, the lack of awareness about how to identify and accommodate different communication needs, and systemic barriers. Strategies to identify patients' communication needs, train healthcare providers, and increase the uptake of communication tools may help improve communication in this population.

---

## Introduction

Conversations between patients, their support network, and their healthcare team are an essential component of high quality healthcare. Good communication is required to set goals of care, provide informed consent, share symptoms, needs, and concerns, discuss and agree on treatment decisions, and exchange information about prognosis or expected outcomes. Poor communication between patients and healthcare teams has been associated with decreased patient satisfaction with care [1], weaker patient-provider relationships [2], increased frequency and duration of hospital stays and emergency department visits [3], reduced patient safety [4], and worse health outcomes [5,6].

People with hearing loss are at risk of experiencing poor communication with healthcare providers and are potentially vulnerable to receiving lower quality or less safe care [7,8]. Patients often note feeling stigmatized and associate hearing loss with negative stereotypes [9,10]. As such, patients may not always feel comfortable disclosing their hearing needs; however, those who do disclose this still report that their needs are often overlooked or disregarded [11], and they are not always provided with the necessary accommodations to communicate [12,13]. Patients with hearing loss also report that they are less likely to be included in conversations about their care, feel less respected by their providers [14], and are more likely to be dissatisfied with patient-provider communication and overall experiences [15]. Certain healthcare settings such as hospitals may be associated with additional communication barriers (e.g., masks, background noise [16]), and the consequences of poor communication may be greater due to the higher acuity and complexity [15,17] of care.

Tools and strategies exist to facilitate communication, such as voice amplifiers and assistive listening devices, hearing devices (e.g., hearing aids, cochlear implants, etc.), captioning, icon boards, body positioning, and lipreading [18]. However, their uptake in the hospital setting is low, despite a healthcare commitment to person-centered care and legislation such as the Accessible Canada Act [19]. This study describes the communication-related experiences of patients with hearing loss in Alberta hospitals. Our goal was to better understand the experiences that this population faces when receiving hospital care, aiming to identify future areas of change.

## Methods

### Study design and setting

This was a qualitative descriptive [20,21] study using focus groups and interviews. We focused on medical or surgical inpatient care at acute care hospitals in Alberta.

### Participants and recruitment

We invited adults living in Alberta that self-reported both hearing loss and being hospitalized within an Alberta hospital in the last 5 years to participate. Support persons were also invited. Participants were recruited between June 13, 2023 and February 14, 2024. We used convenience sampling to recruit participants through posters distributed electronically through non-profit organizations throughout Alberta or approached directly while receiving hospital care in Calgary. Participants were not known to the researchers prior to the study. We recruited until we reached saturation of both demographics (i.e., a diverse sample based on participant demographics including age, self-reported gender, place of residence, primary spoken language, and self-reported socio-demographics) and participant experiences (i.e., no new ideas were presented) [22]. To achieve demographic saturation, we reviewed the demographic characteristics of our participants following completion of the focus groups and purposively recruited additional people for individual interviews if key demographic gaps were found [23,24]. To achieve saturation of participant experiences, analyses were conducted concurrently with recruitment, and we continued to recruit until we felt no new themes were emerging. Complete data analysis details are described below.

Of consenting participants, one withdrew from the study and one was lost to follow up; both had been recruited directly in hospital (one participant requested to withdraw during the interview, and one participant was unable to be reached by telephone to complete the interview). One support person participated in a focus group; however, due to the small sample of support persons, their contributions were not included in our analysis.

This study was approved by the University of Calgary Conjoint Health Research Ethics Board and all participants provided written informed consent.

### Data collection and analysis

Prior to the focus groups or interviews, participants were asked to complete a brief questionnaire related to their self-reported socio-demographics, hearing level, and hospital utilization. Both in-person and online focus groups were offered. In-person focus groups were held at partner community organization facilities in Edmonton and Calgary. The virtual focus group was facilitated via Microsoft Teams. Real-time captioning was provided both in-person and virtually by a live Communication Access Real Time (CART) captioner.

Focus group discussions and individual interviews were facilitated using semi-structured guides (S1 File), developed by members of the research team (ML, AD, SR, NV) based on hearing loss and healthcare communication literature and personal experience of patient partners. To explore patient communication experiences, we designed the guide around an existing communication framework [25] that proposed 4 key factors that can impact communication. To identify potential tools and strategies, we asked questions in alignment with the Alberta Health Services (AHS) Communication Access 6 Strategies framework [26].

Focus groups were co-led by a research associate (ML) and a patient partner (SR). Interviews were conducted by a research coordinator (AD) either in-person at the patient's bedside or over the phone based on participant preference. Field notes were taken by a research coordinator (AD) during both focus groups and interviews to document contextual information not captured in the discussions. Non-participants who were present at the time of the focus groups included CART captioners and representatives from the partnering organization where the focus groups were held. Comments from non-participants were not included in our analyses. No non-participants were present during the individual interviews. CART captioning was provided during focus groups and was also available for interviews if requested. All focus groups

and interviews were audio recorded and transcribed verbatim by a professional transcriptionist experienced in qualitative research transcribing, and de-identified transcripts were uploaded into NVivo14 to facilitate data management and coding.

We implemented a conventional content analysis approach using an abductive thematic coding method to analyze the data [27]. First, we deductively coded the data using the Feldman-Stewart Framework for Communication [17]. We also coded the data using the AHS Communication Access 6 Strategies for Communication [26], a guide developed by allied health professionals and communication specialists to help health professionals communicate with patients with speech, language, or hearing difficulties. We then inductively added new codes outside of these frameworks as directed by the knowledge and expertise of participants. After the focus groups were completed, participant characteristics and field notes were reviewed to ensure diverse perspectives were included and that no new ideas emerged. This analysis occurred while the additional interviews were conducted.

Each transcript was coded independently by two coders trained in qualitative analysis including a research coordinator (AD) and a patient partner (NV) who met regularly to review emergent codes and discuss discrepancies. Other members of the research team (ML, MD) and another patient partner (SR) joined consensus meetings to address discrepancies. Coding occurred iteratively until consensus was reached. We also produced a summarized list of findings and invited participants to provide feedback across two virtual reflective sessions or via email based on participant preference. Reflective sessions lasted approximately one hour and were co-led by a research coordinator (AD) and two patient partners (SR, NV). Live captioning was provided by a CART captioner. Reflective sessions were conducted pre-analysis and served to help us interpret results and ensure initial findings were consistent with participant experiences. When all transcripts were coded and collated, the coders met to discuss and produce a final list of themes.

### Reflexivity

This study was conducted by a multidisciplinary team of academic researchers, clinicians, allied health professionals with diverse backgrounds in health services research and clinical service, and in partnership with community organizations focused on hearing loss. The Principal Investigator (MT) is a male nephrologist and researcher with a special interest in supporting patients with hearing loss. ML contributed to study design and analysis and facilitated focus group discussions. She has a PhD in Health Services Research and experienced in qualitative research. AD took field notes during the discussions, individually interviewed participants, and analyzed the data. She is a Research Coordinator with an MSc in Psychology and was trained on how to take field notes, code transcripts, and interview participants prior to the start of the study by ML. Throughout all stages of this project, we also collaborated with two patient partners (SR, NV); both are women with lived experience of hearing loss, hearing device use (SR wears a cochlear implant, NV wears hearing aids), and hospital encounters. NV and AD coded the transcripts; NV has received qualitative research training through the University of Calgary's Patient and Community Engagement Research (PaCER) program.

Our analyses used a combination of validated frameworks and participant experiences, allowing for a mix of deductive findings based on communication literature and inductive findings based on participant expertise and experience. Findings were reviewed by the larger study team and community organizations to help ensure our findings were accurate reflections of the data and that our language and interpretations were meaningful to Alberta's hearing loss community.

### Results

Fourteen people participated in this study. We ran three focus groups of approximately two hours, consisting of 3–5 participants each (total 11 participants), and three individual interviews of approximately one hour each. After completing the three focus groups, we reviewed participant demographics to evaluate the extent to which the characteristics of the research participants reflect the overall target population (i.e., people with hearing loss in hospital). Because our target population is broad, we focused on achieving representation across certain key characteristics. In particular, we noted that focus group participants were predominantly comprised of participants who identify as women (n = 8). Because gender is

important for both hearing loss and communication in health care [28], we therefore purposively invited participants who identify as men to participate in subsequent individual interviews to achieve a more representative sample. Three men participated in individual interviews.

Of the combined 14 participants, most lived in an urban community. Participants had a range of comorbidities including difficulties with vision, communication, reading, mobility, memory, and difficulty with technology. Most participants reported they did not have difficulty paying for their basic needs (e.g., groceries, housing). All participants had healthcare coverage, and most felt their insurance was sufficient to cover their healthcare needs. All participants used a hearing device, including hearing aids, cochlear implants, bone anchored hearing devices, or a combination of devices. Finally, half of the participants (n = 7) had a hospital experience within the past 6 months, three of whom were receiving hospital care at the time of the interview. The remaining participants' most recent hospital experiences ranged from between 6 months and 1 year (n = 3) to 2–3 years (n = 4) prior to participation. Participant characteristics are reported in Table 1.

**Table 1. Participant characteristics.**

|  | N (%) |
|---|---|
| **Demographic characteristics** | |
| *Self-identified gender* | |
| Men | 6 (43) |
| Women | 8 (57) |
| *Age (mean)* | 30–84 (67) |
| *Geographic location* | |
| Urban | 12 (86) |
| Rural | 2 (14) |
| *Living arrangement* | |
| Lives alone | 5 (36) |
| Lives with others | 8 (57) |
| Residential facility | 1 (7) |
| *Comorbidities* | |
| Vision difficulty | 6 (43) |
| Communication difficulty | 5 (36) |
| Reading difficulty | 2 (14) |
| Mobility difficulty | 2 (14) |
| Memory difficulty | 3 (21) |
| *Technology difficulty* | 4 (29) |
| *Socioeconomic status* | |
| Difficulty paying for needs | 2 (14) |
| No difficulty paying for needs | 12 (86) |
| Sufficient insurance | 8 (57) |
| Insufficient insurance | 4 (29) |
| **Hearing devices worn by participants** | |
| Hearing aid | 10 (71) |
| Bone anchored hearing device | 2 (14) |
| Cochlear implant | 3 (21) |
| Hearing aid & cochlear implant | 1 (7) |
| **Time since last admitted to hospital** | |
| Within the past 6 months | 7 (50%) |
| 6 months to 1 year ago | 3 (21%) |
| 2–3 years ago | 4 (29%) |

From focus group and interview data, we identified three overarching themes: (1) hearing loss is an invisible disability; (2) communication is a team effort; and (3) every patient has different needs in different situations (Fig 1). Additional participant quotes to support each theme and subtheme are presented in S2 Table.

Overall, participants felt that hospitals in Alberta do not meet the communication needs of patients with hearing loss. Three key themes were identified: 1) Hearing loss is an invisible disability: it is often overlooked in healthcare as communication needs are not always apparent, and communication should be equitable; 2) Communication is a team effort: patients must be encouraged and supported to disclose their communication preferences and providers need to accommodate; and 3) Every patient has different needs in different situations: communication tools and strategies can vary based on a patient's individual needs, preferences, and circumstances.

### Theme 1: Hearing loss is an invisible disability

**Subtheme 1: Hearing loss is often overlooked in healthcare.** Hearing loss may not be easy for others to identify. This often results in healthcare providers overlooking it or incorrectly assuming that communication issues are due to cognitive deficits or language barriers. Healthcare providers often are unaware that a person has a hearing difficulty, and most don't ask or know how to identify it. In the focus groups and interviews, the participants clearly described their experiences of hearing loss as being "invisible" and the consequences of this.

*"Hearing [loss] is like an invisible disease because nobody sees it."* (Participant 13)

*"I'm not stupid, I just can't hear properly."* (Participant 14)

*"Certainly it's not on [medical] charts, it's not on your day-to-day care plan, and it's not even a priority because most hearing loss is disregarded and misdiagnosed."* (Participant 4)

**Subtheme 2: Communication should be equitable to all patients.** Being able to participate in conversations with healthcare providers is a right and necessity for high quality care. Participants noted that it can be easier to get accommodations for mobility, vision, or language difficulties, but there are limited options for, or understanding about, hearing support in healthcare.

*"Not being included in your own healthcare can be very detrimental, not only to your mental health but also your physical health, and it's your body, you should be entitled to be fully included, whether you have hearing challenges or not."* (Participant 4)

**Fig 1. Themes identified in focus groups and interviews.**

*"I'm just thinking about the wheelchairs and the mobility and that [because] mobility has huge support in a hospital, so would maybe be something to have a stronger approach to hearing loss. Somebody has worked hard on mobility, maybe it is the people themselves in wheelchairs, I don't know, but, yeah, there is just this general awareness of mobility in hospitals but not of hearing loss."* (Participant 6)

Some participants felt judged by healthcare providers or seen to be causing inconvenience when requesting accommodation.

*"It's about respect and non-judgment on the part of the healthcare professional. I have had times where it almost feels like the person is getting frustrated because I can't hear what they are saying and it is not like it's my fault that I [have hearing loss] … so being able to go to the hospital … and know that I would be accommodated, my needs would be accommodated … having that comfort in knowing I would be treated with respect and have that accommodation that I needed."* (Participant 4)

**Theme 2: Communication is a team effort**

**Subtheme 1: Patients must be encouraged and supported to disclose their communication needs.**  Participants agreed that healthcare providers need to know when someone has a communication need to be able to respond accordingly. As such, participants discussed the importance of patients disclosing their hearing loss and needs.

*"How is a physician or a nurse supposed to know that you can't hear? And you can't expect them to … they change every 12 hours."* (Participant 3)

*"I just look at people and say, look, I am completely deaf, I need your help, and often people will try to accommodate by writing something down for me if. I'm not ashamed to ask for help … having a disability is new to me and I just sort of say it like it is."* (Participant 11)

However, stigma and denial are significant barriers to patients being able to advocate for their needs. More work is needed to help patients feel comfortable disclosing their communication needs.

*"Many people with hearing loss will not admit it. And I don't know how you get around that. It is just a barrier that is there, and there is nothing that the hospital can do to overcome."* (Participant 3)

*"I think there is denial or stigma … I think there are a lot of people who want [hearing loss] confidential and so a lot of [communication strategies] wouldn't work because we have to go through that stage of getting over ourselves and saying, yes, I have this hearing loss, and I need some help."* (Participant 6)

*"I have been an advocate for a long time but I feel especially in healthcare I'm more self-conscious about asking for that support from the receptionist or from a nurse, but I am also part of the LGBTQ2S+ community, and I know that when I see signs … in a pharmacy or doctor's office, I feel safe in that space. Maybe even just a disability pride flag, maybe it is that simple."* (Participant 8)

**Subtheme 2: Providers need to accommodate.**  Participants explained that healthcare providers need to be trained to recognize hearing needs and respond, even if patients don't acknowledge it.

*"There needs to be sensitivity training for the staff so that when they come across people that are having trouble communicating, they are better equipped to deal with the person that is having the communication problems."* (Participant 3).

*"When you live in the hearing loss community you can spot people a mile away who can't hear even though they think they are acting very normally, so perhaps that's the awareness of hospital personnel to recognize that hearing loss."* (Participant 6)

Once communication preferences are identified, participants pointed out that these should be documented in patients' medical charts so that other members of the healthcare team are aware.

*"If [hearing loss] could be incorporated into the paperwork somehow and get passed on to your treatment team … make sure this patient understands what you are saying because there is a hearing [loss] here."* (Participant 14)

Participants also discussed the importance of having communication tools and strategies available and accessible to the healthcare team.

*"The healthcare team carried around little white boards with markers … my family did a voice-to-text on their iPhones so I was able to communicate that way and I found everybody very amenable to that."* (Participant 11)

*"I went into the hospital … I asked 'do you have a Pocketalker?' and they proceeded to say 'Yes, we do … that's in the speech area … [it's] closed and locked up. We can't access that until tomorrow morning'. Well, my appointment is now, so that's not going to help me. So I think whatever is put in place for accommodations needs to be accessible at any time and no matter what department or area you are going to."* (Participant 10)

However, participants noted that healthcare providers are often busy and may not be equipped with the tools and opportunities to ensure good communication. Therefore, the healthcare system plays a key role, including funding for communication tools, training providers, resourcing staff accordingly, and developing policies that facilitate patient-provider communication.

*"It is not only a one-way street for the hard of hearing that need the help. The medical professional part need[s] the help too, and that may be in more ways than I would like to elaborate but it would make things go so much smoother and save a lot of funding in my opinion."* (Participant 10)

*"[Change] comes with administration awareness, locally as well as provincially. The technology is there … So you need to make awareness to the powers that be and that only comes with numbers and lots of noise."* (Participant 5)

**Theme 3: Every patient has different needs in different situations**

**Subtheme 1: People have different needs and preferences for communication solutions.** Hearing loss can present differently for different people and similarly, communication preferences vary. Differences in types of hearing loss, capacity for self-advocacy, comorbidity, technology comfort, and other patient preferences can all influence communication, patient experiences, and outcomes.

*"Even knowing things like there are different types of hearing loss and different things work for other people … when people speak really loud, that's great for me [but] most people don't want you to speak loud. That's not helpful to them … And knowing that just because something worked for one person may not work for someone else it would be useful…"* (Participant 2)

Given this variation in hearing loss presentation and support preference, participants felt that a variety of communication tools and strategies should be available to support individual needs and preferences, guided by patient preference.

*"I think that if an organization has multiple approaches available, like say they might have a hearing induction loop at many points of the hospital … maybe they have a Pocketalker that can be borrowed to go around the hospital, perhaps they have some sort of device that can caption … it [addresses] multiple disabilities, different challenges. So I think it is really important that they take a multiple [approach]."* (Participant 10)

One participant spoke to problems using tools such as speech-to-text captioning due to vision difficulties, highlighting the necessity for a variety of tools and the need to be mindful of coexisting disabilities and patient abilities.

*"Captioning I'm sure is really great for people who only [have hearing loss] but for me, somebody who [has vision loss], captioning doesn't work unless you make it size 500 and give me like 2 minutes to read one sentence, you know. It doesn't work for me, and I don't know how we make it more accessible for people who [are] also visually impaired."* (Participant 8)

**Subtheme 2: What works in one setting may not work in others.** In addition to people having varying preferences, different situations might impact which tools are best. For example, hearing induction systems (i.e., loops) may be useful, but many facilities lack the infrastructure to support them.

*"It may have T-coils, but if the audiologist or practitioner doesn't turn them on and set them up, it won't work. Just because a hearing aid has a T-coil in it doesn't mean it is always going to work."* (Participant 3)

Other tools may be useful for simple conversations, but not those with complicated language. Therefore, understanding the context and setting was thought to be needed to help dictate which tools and strategies are most appropriate.

*"They would get a piece of paper. But you're not going to get that real detailed discussion that you need to hear at the moment where they may tell you a lot of information and they will just give the surface stuff because I can't hear. I'm just getting the basics."* (Participant 1)

*"The thing about the [captioning] apps is they're not going to be able to know the name of medication or anything that is very specialized. So, they become a lot less useful the more in depth the conversation is."* (Participant 2)

## Discussion

This study described the communication experiences of adult patients with hearing loss while in hospital. Three main themes emerged: (1) hearing loss is an invisible disability, (2) communication is a team effort, and (3) every patient has different needs in different situations. Our findings align with previous research aimed at determining the impacts of hearing loss on care, as follows.

### Hearing loss is an invisible disability

Because hearing loss is not always apparent to others, challenges with communication may go unnoticed, and hence people with hearing loss are often under-supported. In some circumstances, patients may advocate for themselves by letting their healthcare providers know they have hearing loss, but others do not due to stigma or worries of causing inconvenience [29]. In our study, some participants perceived that it may be easier to access supports for physical disabilities than for communication needs. Others have also found that compared to those with physical disabilities, hearing-related needs were perceived to be often overlooked and unaccommodated [30]. Additionally, people with hearing loss often experience perceived or real stigmatizing attitudes in the form of ageism or ableism [11]. Thus, it is suggested that efforts are needed

at the provider, institutional, and societal level to change the culture in healthcare, emphasizing each patient's right to effective communication.

In addition, stigma-reduction interventions are critical to help create equitable access to health and other services. First, public health and advocacy organizations may help reduce stigma by increasing the visibility of hearing loss. For example, the World Health Organization (WHO) promotes World Hearing Day each year on March 3 and distributes educational and advocacy materials broadly [31]. Second, helping healthcare providers become aware of their attitudes and behaviour may help reduce stigma in patients with hearing loss. For example, a scale was recently developed and validated [32] to help providers become aware of their own stigmatizing attitudes and identify areas for change. Finally, encouraging people with hearing loss to feel empowered and advocate for themselves can help reduce feelings of stigmatization. While some may avoid help seeking or attempt to mask their condition by physically hiding hearing devices or pretending to understand what is being said [29,30], others have noted that using humor and maintaining a positive attitude, ignoring or confronting stigmatizing attitudes or people, self-advocacy, educating others about hearing loss, and learning to accept their hearing loss can effectively reduce stigma [30].

## Communication is a team effort

Participants in our study felt that patients (with hearing and communication supports), healthcare providers, and the healthcare system all play a role in ensuring good communication. While providers and the healthcare system have a responsibility to provide the tools, training, and policies to accommodate, patients should also be encouraged to inform their providers of their communication needs.

However, people do not always feel comfortable acknowledging their hearing challenges [29]. One study of adult inpatients at a urban hospital found that 72.8% had audiometric hearing loss; however, only 28.8% of these patients self-reported having difficulty hearing physicians and nurses [33]. Reasons for this discrepancy could include fears of stigmatization or discrimination [11], perceptions that hearing loss is not as important as other medical concerns [29], or influenced by individual patient needs and circumstances as discussed below. In addition, other research has also found that patients feared creating an additional burden for healthcare staff, and hearing loss was only disclosed if patients' felt their provider showed interest, care, and concern about their communication difficulties [34]. Similar sentiments were also shared by participants in another qualitative study, who noted that familiarity and comfort with who they were speaking with influenced how likely they were to disclose their hearing loss [29].

Therefore, we suggest that healthcare providers be equipped with the skills, responsibility, and confidence to identify patients with hearing loss and communicate effectively. In our study, many participants called for improved sensitivity training, and noted that simple strategies such as being patient, empathetic, and respectful would help to improve communication.

Training healthcare providers on how to recognize hearing loss and understand the impact it may have on health and communication may help providers become better equipped to accommodate communication needs. Previous qualitative work has found that healthcare providers' lack of awareness about hearing loss and patients' communication needs were barriers to care [35]. Survey data has further revealed that only 3.5% of healthcare providers received formal training about how to communicate with patients with hearing loss, thus amplifying the urgency of formal training to understand and address the communication needs of this population, as well as familiarity with assistive technologies [36]. Finally, a participatory action study [37] of capacity building among healthcare providers to support communication with Deaf and hard of hearing patients found providers often felt ill-equipped to support these patients, and suggested that training for healthcare providers should be co-developed and led by patients with hearing loss, and include awareness of hearing loss and how it impacts patients, communication strategies such as speaking slowly and clearly, scenario and role-playing exercises, and improved awareness of and easy access to existing communication resources. Ultimately, this study found that this capacity building intervention was effective in increasing self-efficacy among healthcare providers.

Staff capacity-building initiatives have also been successful in supporting communication for patients with other communication needs such as aphasia. In one study of healthcare providers working on a stroke rehabilitation unit, providers helped to develop guidelines, a training course, and communication tools to support communication with aphasia patients. Through these practices, providers showed an increased understanding and reduced frustration during communication, and felt confident in using communication techniques such as slower speech, facing the patient, use of written or picture information, and allowing extra time for communication [38].

Other institutional policies and procedures such as documenting the presence of hearing loss in the medical record may help inform and remind providers about a patient's communication needs and preferred accommodations [39], without relying on patients to repeatedly disclose their condition. Participants in our study indicated that communication tools (e.g., assisted listening devices or real time captioning) were sometimes available in theory; however, physical or policy barriers such as a tool being locked up or in use elsewhere often reduced their accessibility when needed. Thus, we suggest that providing healthcare providers with a variety of communication tools and strategies that are *easily accessible* [40] will help patients communicate in a way that is most comfortable for them.

## Every patient has different needs in different situations

Participants expressed that there is great variety in needs and preferences for communication supports. Therefore, different tools and strategies should be available to accommodate each person in a full range of clinical situations. A recent scoping review identified several communication strategies that can be used by healthcare providers that can be applied to a variety of situations and patient needs and preferences [41].

The clinical setting in which patients with hearing loss are receiving care may also impact which communication tool or strategy is most appropriate. In one-on-one appointments, private rooms, or reasonably quiet environments, strategies such as increasing speech volume, body language adjustments, or captioning may be helpful. However, settings such as a busy emergency department where patients are not able to communicate in a one-on-one setting and there is excessive background noise will require other strategies. Patients with hearing loss also may have trouble hearing their name called in busy waiting rooms [42], so strategies such as written or visual cues to indicate when their name has been called are helpful. In addition, while one study found that providing sound amplifiers to patients with hearing loss in the emergency room helpful in facilitating communication [43], many emergency departments may not be equipped to do so. Finally, patients also report difficulty hearing over the phone [12], so in-person care, virtual meetings, or written information such as text or email-based information appointment reminders may be preferable.

Through qualitative discussions with patients with hearing loss, we propose simple strategies to better meet the needs of patients with hearing loss, with opportunities for improvement existing at the patient, provider, and healthcare system levels [44]. Patients should be encouraged and supported to disclose their hearing challenges and advocate for what they need; however, such disclosure must be reciprocated by commensurate supports. Additional training may be helpful in supporting healthcare providers to better identify hearing loss-related communication challenges and accommodate patients' needs. In addition, communication tools should be easily available, and processes to identify and document communication needs within the medical record should be implemented. Initiatives by local groups such as the Communication Access team within Alberta Health Services are currently working to improve communication for patients with communication difficulties by providing publicly available tools and resources and training modules for healthcare providers. These initiatives may help address all three barriers identified in this report by increasing the visibility of communication needs, enabling healthcare providers with the skills and knowledge to accommodate patients with communication needs, and encouraging facilities to invest in a variety of tools and strategies to meet the diversity in patient needs. These efforts and relatively simple changes proposed in this report would be expected to promote meaningful patient participation in health care, thereby improving patient empowerment, shared decision making, and improved satisfaction with care [45].

## Study strengths and limitations

This study has many strengths. We included a diverse sample of people with hearing loss from across Alberta with a range of demographics, hospital experiences, types of hearing loss, and hearing device use. We used a comprehensive and community-based recruitment process and provided captioning by a CART professional to ensure participation in the discussions was accessible. The group interaction of a focus group allowed participants to have a rich conversation and build ideas from one another. Discussions were transcribed verbatim, and coding was based on validated and established frameworks. Finally, this work was completed amongst a collaborative study team including academic researchers, clinicians, health system leaders, and patient partners who have lived experience with hearing loss.

This study also has limitations to consider. First, we focused on the experience of Albertans with hearing loss while in the hospital, and so findings may not reflect the experiences of those in other settings. Second, we asked participants to reflect on a prior hospital encounter (up to 2–3 years prior to the focus group or interview) which may have limited accuracy of shared experiences due to recall bias or memory distortion. Third, only two participants lived in a rural location, and no one with a significant language difficulty or had low socioeconomic status participated, so the findings and suggestions we propose in this report may not fully reflect the experiences of these groups. Lastly, we excluded people who identified solely as Deaf, because the communication tools and strategies to support those who are hard of hearing likely differ from those that would benefit people in the Deaf community. As such, our findings may not be fully generalizable to the broader Deaf and hard of hearing community.

## Conclusion

This qualitative descriptive study explored the communication-related experiences of patients with hearing loss in Alberta hospitals. We identified 3 key themes: hearing loss is an invisible disability, communication is a team effort, and every patient has different needs in different situations. Three key strategies may help improve communication in hospital. First, increasing the awareness of hearing loss both societally and among healthcare workers may help reduce stigma, allowing patients to feel more comfortable disclosing their needs. Second, encouraging change at the patient, healthcare provider, and health system level can help ensure patient needs are identified and accommodated. Finally, ensuring that providers are trained and have access to a variety of communication tools and strategies will help accommodate the full range of patient preferences.

## Supporting information

**S1 File. Focus group and interview guides.**
(DOCX)

**S2 Table. Additional participant quotes to support each theme and subtheme.**
(DOCX)

## Acknowledgments

The authors of this report are grateful to Laura Slywka and Karen Munro for providing captioning services during the focus group discussions. We also thank the Deaf and Hear Alberta (DHA) organization and the Calgary and Edmonton branches of the Canadian Hard of Hearing Association (CHHA) for their support with participant recruitment, hosting the focus groups, and ongoing collaboration.

## Author contributions

**Conceptualization:** Meaghan Lunney, Marcello Tonelli.

**Data curation:** Alex DeBusschere, Meaghan Lunney, Sonja Reid.

**Formal analysis:** Alex DeBusschere, Meaghan Lunney, Sonja Reid, Nancy Verdin.

**Funding acquisition:** Marcello Tonelli.

**Project administration:** Alex DeBusschere, Meaghan Lunney.

**Supervision:** Meaghan Lunney, Marcello Tonelli.

**Writing – original draft:** Alex DeBusschere, Meaghan Lunney.

**Writing – review & editing:** Sonja Reid, Nancy Verdin, Shannan Love, Stephanie Thompson, David Nicholas, Tiffany Boulton, Kara Schick Makaroff, Lorienne Jenstad, Sharon Straus, Jayna Holdroy-Leduc, Maoliosa Donald, Patti-Jo Sullivan, Tanis Howarth, Julie Evans, Marcello Tonelli.

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
