## [Decision Letter · Decision Letter 0]

24 Jul 2025

Dear Dr. Tonelli,

Thank you for submitting your manuscript to PLOS ONE. After careful consideration, we feel that it has merit but does not fully meet PLOS ONE’s publication criteria as it currently stands. Therefore, we invite you to submit a revised version of the manuscript that addresses the points raised during the review process.

All the changes suggested by reviewers have merit and must be considered.  Please use COREQ guidelines to improve the reporting of your qualitative research.I would recommend that you consider suggestions regarding patient characteristics and reflexivity  as important for publication.If there are any conflicts in feedback, know that whatever you change may influence the logical flow of the article. Keep your COREQ reporting guidelines in mind when in doubt. Your research will create awareness in health care for a vulnerable population. 

We look forward to receiving your revised manuscript.

Kind regards,

Deidre Pretorius, PhD

Academic Editor

PLOS ONE

Journal Requirements:

[The study was supported by a CIHR project grant to MT (award number 202309PJT). The funding organization had no role in the design and conduct of the study; collection, management, analysis, and interpretation of the data; preparation, review, or approval of the manuscript; and decision to submit the manuscript for publication.].

4. Please amend the manuscript submission data (via Edit Submission) to include author Alex DeBusschere.

5. Please amend your authorship list in your manuscript file to include author Alexandra DeBusschere.

Additional Editor Comments:

Dear Author(s)

Please see the feedback from reviewers. We would like to see the revised article.

REgards

D

Reviewers' comments:

Reviewer's Responses to Questions

**Comments to the Author**

1. Is the manuscript technically sound, and do the data support the conclusions?

Reviewer #1: Yes

Reviewer #2: Yes

Reviewer #3: Partly

2. Has the statistical analysis been performed appropriately and rigorously?

Reviewer #1: N/A

Reviewer #2: N/A

Reviewer #3: Yes

3. Have the authors made all data underlying the findings in their manuscript fully available?

Reviewer #1: No

Reviewer #2: No

Reviewer #3: No

4. Is the manuscript presented in an intelligible fashion and written in standard English?

Reviewer #1: Yes

Reviewer #2: Yes

Reviewer #3: Yes

Reviewer #1: This study addresses an important topic and has been carried out to a high standard. The manuscript is written and presented well.

The one aspect of this study that was unclear to me was the basis for ‘saturation’ in the demographic data. The lack of ‘saturation’ in the demographics of the focus group data was the reason why 3 interviews were added to the data from the focus group (this step is somewhat unorthodox). Did you consider the representativeness of your sample relative to the population of people who live in Alberta and with hearing loss?

N.B. I have responded 'no' to the reviewer question about making the data fully available. I support the authors' decision not to make the data from these qualtiative interviews and focus groups fully available, as they may contain sensitive information.

** Introduction **

Line 77. “Initiatives by local groups such as the Communication Access team…”. Please provide some examples of these initiatives.

** Methods **

Line 94. “…until we reached saturations of both demographics..”. What method was used to establish that “saturation” had been reached for the demographics?

Line 116. How were the guides for the focus groups and the interviews developed? Were they piloted before data collection took place?

Lines 120-122. Were any non-participants present during the interviews?

Lines 122-124. Were any field notes made during the focus groups/interviews?

Line 137. “…by two coders trained in qualitative analysis…”. What kind of training have these two coders attended?

Line 139. Are you able to provide a measure of the level of agreement between these two coders? How were discrepancies resolved?

** Results **

Table 2 provides a very comprehensive account of the minor themes supported by relevant quotes from a wide range of participants. Perhaps Table 2 could appear in Supplementary Materials as there is some redundancy in the write up of the results i.e. some of the same participant quotes appear in Table 2 and the written description of the themes and subthemes.

** General **

Please use ‘high quality care’ or ‘high-quality care’ throughout.

Reviewer #2: Thank you for sharing your manuscript, “Hospital care does not meet the communication needs of patients with hearing loss: A qualitative study of patient experiences.” I appreciate the opportunity to review such a thoughtful and timely contribution.

Strengths of the study

• Your study is a much-needed exploration of a topic that impacts many patients and their families, yet is rarely addressed in hospital policy or training.

• The qualitative approach is very well suited for the research questions and gives a clear, real-world voice to participants’ needs and insights.

• The reporting is meticulous, with methods, recruitment, and analysis explained step-by-step and transparently.

• The use of direct quotes throughout the results brings authenticity and clarity.

Where You Can Improve

• I appreciate that you have addressed both demographic and experiential saturation and listed the key variables considered. If possible, adding a brief description of how you monitored for saturation (for example, indicating after which interviews or groups you judged that saturation was reached) would further strengthen clarity and transparency for the reader.

• Since participants reflected on hospital experiences from the past several years, you might clarify what steps, if any, were taken to reduce the possible influence of recall bias or memory gaps.

• The manuscript is clear overall, just a few minor issues with repeated wording and phrasing could be rephrased for smoother reading. For example, in the abstract, the phrase 'patients with hearing loss' is used repeatedly in close proximity:

“Tools and strategies to facilitate two-way communication with patients with hearing loss and other members of their healthcare team may help address this gap. This study describes the communication-related experiences of patients with hearing loss in Alberta hospitals, which can help inform future strategies in this setting.”

Rewording for conciseness, perhaps by substituting with 'these patients' or streamlining sentences, would make for smoother reading.

I also noticed similar patterns in the results and introduction, where closely related ideas are covered in multiple, slightly different ways. Consolidating or varying phrasing in these sections would further enhance clarity and flow.

• I encourage you to be mindful in your discussion section that practice and policy suggestions are framed as proposed improvements, not prescriptive recommendations. This preserves both the strength and boundaries of qualitative findings.

• You have clearly explained why data can’t be shared openly. If feasible within your institution, stating whether access is possible for qualified researchers by request could further support transparency.

Thank you again for your dedication to equity in healthcare communication. This research will be invaluable for clinicians, administrators, and advocates seeking to improve hospital experiences for patients with hearing loss.

Reviewer #3: Overall Impression

This is an insightful and timely article, and the research undertaken represents an important contribution to the field, offering valuable insights that could significantly enhance patient care.

Recommendations for Revision and Clarification

• Line 168: Missing Figure Reference. Line 168 refers to "Figure 1"; however, no figure with this designation appears within the manuscript. Please ensure that all cited figures are present and correctly numbered, or remove the reference if the figure is not intended for inclusion.

• Table 1: Participant Characteristics. The current presentation of participant characteristics in Table 1, while providing aggregate data, limits the reader's ability to connect specific participant attributes with their quoted contributions in the results and discussion. To enhance clarity and facilitate deeper understanding, I recommend restructuring this table to present individual participant profiles. An example of a more effective table layout, which would better support the analysis and discussion of qualitative data by allowing direct linking of quotes to specific participant demographics and experiences, has been provided separately.

• Table 2: Redundant Information. Table 2 appears to contain information that is duplicated or more effectively integrated within the narrative of the results and discussion sections. To streamline the manuscript and improve readability, consider removing this table. The pertinent quotes and findings currently presented in Table 2 should be directly incorporated into the body of the text, serving to illustrate and substantiate the themes discussed in the respective sections.

• Interviews Conducted: Rationale for Male Participant Interviews. The rationale for conducting in-depth interviews specifically with male participants, as opposed to a focus group approach similar to that used for other participant groups, requires further clarification. Please elaborate on the methodological decision-making process behind this choice. Highlighting the specific advantages of individual interviews for male participants in this context, or explaining any practical constraints that led to this approach, would strengthen the methodological transparency.

• Data Analysis and Discussion: While the results and discussion sections effectively present the study's findings and themes, their analytical depth would be significantly enhanced by explicitly linking these insights to the diverse demographic characteristics of the participants. Integrating individual participant profiles (as suggested for Table 1) directly into the discussion would allow for a more nuanced exploration of how variations in age, gender, geographic location, comorbidities, or living arrangements might shape or influence the reported experiences and perspectives. This approach would not only provide richer context for interpreting quotes but also enable the identification of patterns, exceptions, or differential impacts across various participant groups, thereby strengthening the study's transferability and revealing potential areas for targeted interventions or future research

• Researcher Considerations: Bias and Reflexivity. The inclusion of patient partners in the research process is highly commendable and enhances the rigor and relevance of this study. To further strengthen this section, it is important to explicitly address any potential biases or conflicts of interest that the patient partners may have brought to the interview or focus group settings, or to the analysis phase. Additionally, incorporating a dedicated reflexivity section would significantly benefit the manuscript. This section should detail the researchers' and patient partners' positions relative to the research, how their experiences and perspectives might have influenced data collection and interpretation, and the strategies employed to mitigate potential biases. This inclusion would significantly enhance the academic robustness of the study.

**Do you want your identity to be public for this peer review?** For information about this choice, including consent withdrawal, please see our Privacy Policy

Reviewer #1: No

Reviewer #2: **Yes: ** Carmen Milton

Reviewer #3: No

---

## [Author Response · Author response to Decision Letter 1]

1 Aug 2025

Done.

Please provide an amended funding statement that declares *all* the funding or sources of support (whether external or internal to your organization) received during this study, as detailed online in our guide for authors at http://journals.plos.org/plosone/s/submit-now.  Please also include the statement “There was no additional external funding received for this study.” in your updated Funding Statement. Please include your amended Funding Statement within your cover letter. We will change the online submission form on your behalf.

Done.

We note that you have indicated that there are restrictions to data sharing for this study. For studies involving human research participant data or other sensitive data, we encourage authors to share de-identified or anonymized data. However, when data cannot be publicly shared for ethical reasons, we allow authors to make their data sets available upon request. For information on unacceptable data access restrictions, please see http://journals.plos.org/plosone/s/data-availability#loc-unacceptable-data-access-restrictions. Please update your Data Availability statement in the submission form accordingly.

Done.

Please amend the manuscript submission data (via Edit Submission) to include author Alex DeBusschere.

Done.

Please amend your authorship list in your manuscript file to include author Alexandra DeBusschere.

Done.

Please include captions for your Supporting Information files at the end of your manuscript, and update any in-text citations to match accordingly.

Done.

Reviewer 1:

Methods: The one aspect of this study that was unclear to me was the basis for ‘saturation’ in the demographic data. The lack of ‘saturation’ in the demographics of the focus group data was the reason why 3 interviews were added to the data from the focus group (this step is somewhat unorthodox). Did you consider the representativeness of your sample relative to the population of people who live in Alberta and with hearing loss?

We have revised the Methods and Results section to better clarify how we assessed and achieved saturation of the demographic data. See lines 93-98 and 184-192. Qualitative researchers often use combined data collection methods (such as interviews, focused group discussions, observations, field notes, etc.) to better achieve saturation compared to using a single data collection approach. See Rahimi and Khatooni, Int J Nursing Studies Adv 6 (2024); Lamberta and Loiselle, J Adv Nurs (2008)

Line 77. “Initiatives by local groups such as the Communication Access team…”. Please provide some examples of these initiatives.

We have added some examples. Note, we moved this point to the discussion as it is more relevant in how the healthcare system can address the findings identified in this study. Please see lines 487-494.

Line 94. “…until we reached saturations of both demographics..”. What method was used to establish that “saturation” had been reached for the demographics?

We have added additional information to clarify how we determined if we have reached saturation. As above, please see lines 93-98.

Line 116. How were the guides for the focus groups and the interviews developed? Were they piloted before data collection took place?

We have added additional information about how the guides were developed. Please see lines 116-121.

Lines 120-122. Were any non-participants present during the interviews?

We have reported if non-participants were present during the focus groups and interviews. Please see lines 126-130.

Lines 122-124. Were any field notes made during the focus groups/interviews?

We have noted that field notes were made during the discussions. Please see lines 124-126.

Line 137. “…by two coders trained in qualitative analysis…”. What kind of training have these two coders attended?

We have added an additional Reflexivity section in the methods section of the manuscript that outlines the training that the two coders had received. Please see lines 168-169 and 172-174.

Line 139. Are you able to provide a measure of the level of agreement between these two coders? How were discrepancies resolved?

While we did not quantitatively measure the level of agreement between the two coders (for example, via intercoder reliability [ICR]), we are confident that we have achieved a robust analysis and have demonstrated reliability and trustworthiness through other methods in our analyses including an iterative coding process. In our methods section, we have further clarified how discrepancies were resolved and how agreement was reached. Please see lines 148-150.

Table 2 provides a very comprehensive account of the minor themes supported by relevant quotes from a wide range of participants. Perhaps Table 2 could appear in Supplementary Materials as there is some redundancy in the write up of the results i.e. some of the same participant quotes appear in Table 2 and the written description of the themes and subthemes.

Done.

General: Please use ‘high quality care’ or ‘high-quality care’ throughout.

Done.

Reviewer 2:

If possible, adding a brief description of how you monitored for saturation (for example, indicating after which interviews or groups you judged that saturation was reached) would further strengthen clarity and transparency for the reader.

We have provided additional detail about how demographic and thematic saturation was determined. Please see lines 93-98.

Since participants reflected on hospital experiences from the past several years, you might clarify what steps, if any, were taken to reduce the possible influence of recall bias or memory gaps.

We have added additional details about the length of time participants were recalling experiences from. This has been added to the manuscript text as well as Table 1. The influence of recall bias or memory gaps have also been addressed in the limitations section of the discussion.

The manuscript is clear overall, just a few minor issues with repeated wording and phrasing could be rephrased for smoother reading. Consolidating or varying phrasing in these sections would further enhance clarity and flow.

Done. We have revised the manuscript for clarity and conciseness.

I encourage you to be mindful in your discussion section that practice and policy suggestions are framed as proposed improvements, not prescriptive recommendations. This preserves both the strength and boundaries of qualitative findings.

Done. We have revised the wording in the discussion to frame our findings as suggested improvements.

You have clearly explained why data can’t be shared openly. If feasible within your institution, stating whether access is possible for qualified researchers by request could further support transparency.

We have updated our data access statement to indicate that de-identified data may be made available to qualified researchers upon reasonable request to the corresponding author.

Reviewer 3:

Line 168: Missing Figure Reference. Line 168 refers to "Figure 1"; however, no figure with this designation appears within the manuscript. Please ensure that all cited figures are present and correctly numbered, or remove the reference if the figure is not intended for inclusion.

As per the Journal’s submission guidelines, we have referred to Fig. 1 in the text and included the Figure title and caption, but the figure file has been uploaded as a separate file attachment.

Table 1: Participant Characteristics. The current presentation of participant characteristics in Table 1, while providing aggregate data, limits the reader's ability to connect specific participant attributes with their quoted contributions in the results and discussion. To enhance clarity and facilitate deeper understanding, I recommend restructuring this table to present individual participant profiles. An example of a more effective table layout, which would better support the analysis and discussion of qualitative data by allowing direct linking of quotes to specific participant demographics and experiences, has been provided separately.

Thank you for this suggestion. We agree that including detail on how findings varied based on participant characteristics can provide valuable context about participant experiences; however, we believe that presenting our data in a participant profile format poses risks of potentially identifying participants. As such, we have decided to keep the table as aggregate data.

Table 2: Redundant Information. Table 2 appears to contain information that is duplicated or more effectively integrated within the narrative of the results and discussion sections. To streamline the manuscript and improve readability, consider removing this table. The pertinent quotes and findings currently presented in Table 2 should be directly incorporated into the body of the text, serving to illustrate and substantiate the themes discussed in the respective sections.

We have removed Table 2 from the main text and moved it to the Supporting Information section.

Interviews Conducted: Rationale for Male Participant Interviews. The rationale for conducting in-depth interviews specifically with male participants, as opposed to a focus group approach similar to that used for other participant groups, requires further clarification. Please elaborate on the methodological decision-making process behind this choice. Highlighting the specific advantages of individual interviews for male participants in this context, or explaining any practical constraints that led to this approach, would strengthen the methodological transparency.

Qualitative research often uses a combination of focus groups and individual interviews for pragmatic reasons and to reach data saturation/completeness. In our study, we first conducted focus groups which allow for interactions and discussions among group members. Following the focus groups, we identified key demographic gaps (specifically, most of our participants in the focus groups identified as women). To achieve a more representative sample, we decided to conduct additional individual interviews and recruited participants that identified as male while they were receiving hospital care. This method allowed us to recruit participants from this underrepresented group more flexibly and pragmatically. We have clarified our decision to first conduct focus groups followed by purposive individual interviews and have provided a citation to support the combination of these methods. Please see lines 93-96 and 184-192.

Data Analysis and Discussion: While the results and discussion sections effectively present the study's findings and themes, their analytical depth would be significantly enhanced by explicitly linking these insights to the diverse demographic characteristics of the participants. Integrating individual participant profiles (as suggested for Table 1) directly into the discussion would allow for a more nuanced exploration of how variations in age, gender, geographic location, comorbidities, or living arrangements might shape or influence the reported experiences and perspectives. This approach would not only provide richer context for interpreting quotes but also enable the identification of patterns, exceptions, or differential impacts across various participant groups, thereby strengthening the study's transferability and revealing potential areas for targeted interventions or future research

Thank you for this suggestion. We agree that this would be a useful direction to consider in future research to develop more targeted interventions. As above, to avoid risks of identifying participants, we have not expanded Table 1 to include individual participant characteristics. In addition, as we did not intend to analyse our data based on individual characteristics, we did not purposively recruit to achieve sufficient sample sizes in each sub-group. As such, we are unable to draw conclusions or make comparisons across specific demographics or groups in this way. However, we do refer to examples that were brought forward by participants such as a participant with a comorbid vision difficulty.

Researcher Considerations: Bias and Reflexivity. The inclusion of patient partners in the research process is highly commendable and enhances the rigor and relevance of this study. To further strengthen this section, it is important to explicitly address any potential biases or conflicts of interest that the patient partners may have brought to the interview or focus group settings, or to the analysis phase. Additionally, incorporating a dedicated reflexivity section would significantly benefit the manuscript. This section should detail the researchers' and patient partners' positions relative to the research, how their experiences and perspectives might have influenced data collection and interpretation, and the strategies employed to mitigate potential biases. This inclusion would significantly enhance the academic robustness of the study.

We have added a Reflexivity section in the Methods section that outlines researcher and patient partner roles and backgrounds. Please see lines 159-180.

---

## [Decision Letter · Decision Letter 1]

16 Sep 2025

Hospital care does not meet the communication needs of patients with hearing loss: A qualitative study of patient experiences.

PONE-D-25-24516R1

Dear Dr. Tonelli,

We’re pleased to inform you that your manuscript has been judged scientifically suitable for publication and will be formally accepted for publication once it meets all outstanding technical requirements.

Kind regards,

Laura Kelly, PhD

Division Editor

PLOS One

Reviewers' comments:

Reviewer's Responses to Questions

**Comments to the Author**

Reviewer #1: All comments have been addressed

Reviewer #3: All comments have been addressed

2. Is the manuscript technically sound, and do the data support the conclusions?

Reviewer #1: Yes

Reviewer #3: Yes

3. Has the statistical analysis been performed appropriately and rigorously?

Reviewer #1: N/A

Reviewer #3: Yes

4. Have the authors made all data underlying the findings in their manuscript fully available?

Reviewer #1: No

Reviewer #3: Yes

5. Is the manuscript presented in an intelligible fashion and written in standard English?

Reviewer #1: Yes

Reviewer #3: Yes

Reviewer #1: Thank you for addressing all my comments. This manuscript is very well-written and clear and I look forward to seeing it in press.

Reviewer #3: (No Response)

**Do you want your identity to be public for this peer review?** For information about this choice, including consent withdrawal, please see our Privacy Policy

Reviewer #1: No

Reviewer #3: No

---

## [Editor Report · Acceptance letter]

PONE-D-25-24516R1

PLOS ONE

Dear Dr. Tonelli,

I'm pleased to inform you that your manuscript has been deemed suitable for publication in PLOS ONE. Congratulations! Your manuscript is now being handed over to our production team.

Kind regards,

on behalf of

Dr. Laura Hannah Kelly

Staff Editor

PLOS ONE